# OpenReview forum: "Extending Stability Analysis to Adaptive Optimization Algorithms Using Loss Surface Geometry"
_ICLR.cc/2025/Conference — Submitted to ICLR 2025_

### Official Review · Reviewer_c15z · 2024-10-23

**Soundness:** 1
**Presentation:** 1
**Contribution:** 1
**Rating:** 1
**Confidence:** 3

**Summary:**

This paper provides necessary and sufficient conditions for the linear stability of adaptive optimization methods (such as RMSProp, Adam). The authors introduce a generalized coherence measure to capture the interaction between the adaptive preconditioner and the Hessian, which is then used to derive a linear stability condition. The theoretical findings are supported by experiments conducted on real-world datasets.

**Strengths:**

The paper is easy to follow and tackles the important problem of analyzing stability conditions of adaptive methods, well-motivated by a preliminary experiment comparing the generalization ability of SGD and Adam.

**Weaknesses:**

This paper contains several significant technical flaws, with mathematical arguments that lack both rigor and clarity. Key issues include:

1. **The notion of "linear stability" is not clearly defined**.
   - The concept of linear stability is central to the paper's claims, yet it is never explicitly defined. Since there are a few diffenet flavors of linear stability in the literature, this omission weakens the theoretical foundation. The authors should provide a precise definition before presenting any theorems or formal arguments based on it.

2. **The linear stability condition for SGD is incorrect**.
   - In Section 2.1, the paper presents the condition $\lVert I - \eta H(\theta^*) \rVert < 1$ as the linear stability criterion for SGD (Eq. 2). While this holds for (full-batch) GD, it does not apply to SGD due to the noise introduced by stochastic sampling. As demonstrated by prior works such as [Wu et al., 2018] and [Wu et al., 2022], the stability of SGD also depends on factors such as noise covariance and batch size, none of which are accounted for in the paper's condition.

3. **The stability condition for adaptive methods is incorrectly derived**.
   - In Section 3.3.1, the paper introduces $p_i$ as the diagonal element of $P^*$ corresponding to the eigenvalue $\lambda_i$ of $H(\theta^*)$. This reasoning holds only when the eigenvectors of the Hessian are aligned with the coordinate axes, i.e., when the Hessian is diagonal. However, in general, the eigenvectors are not aligned with the coordinate system, making Eq. (13) and the subsequent analysis incorrect. As this forms the basis of the paper's linear stability condition for adaptive methods, the main theoretical result is flawed.

4. **The proof of convergence for the Adam preconditioner in Appendix B is incorrect**.
   - The paper claims to prove that the Adam preconditioner $P_t$ converges to a constant matrix $P^*$ as $t \to \infty$. However, the proof in Appendix B lacks rigor. In Line 672, the argument that exponential decay of earlier gradients implies that $v_t$ reaches a steady state is unsubstantiated. For such a claim to be valid, the assumptions must be explicitly stated, and the proof must be strengthened with a more rigorous analysis.

Additionally, the paper omits citation of a relevant work, [Cohen et al., 2022], which analyzed the stability condition of adaptive methods (assuming stationary preconditioners) on quadratic problems. According to their results, the stability condition for RMSProp (and Adam) is that the preconditioned sharpness $\lambda_{\max}(P^{-1}H)$ remains below $2/\eta$ ($38/\eta$ for Adam). This paper should cite [Cohen et al., 2022] and provide a careful discussion comparing and contrasting their results with the findings in this work.

---

**References**

[Wu et al., 2018] How SGD Selects the Global Minima in Over-parameterized Learning: A Dynamical Stability Perspective, NeurIPS 2018.

[Wu et al., 2022] The alignment property of SGD noise and how it helps select flat minima: A stability analysis, NeurIPS 2022.

[Cohen et al., 2022] Adaptive Gradient Methods at the Edge of Stability, arXiv preprint 2022.

**Questions:**

Could you provide more details on the experimental setup for Tables 1 and 2? Specifically, what batch size and number of epochs were used, and were the hyperparameters carefully tuned for each setting? Additionally, how was sharpness computed in the experiments according to Eq. (24)?

---

### Official Review · Reviewer_PX69 · 2024-10-24

**Soundness:** 1
**Presentation:** 2
**Contribution:** 1
**Rating:** 3
**Confidence:** 4

**Summary:**

In this work, the authors extend the linear stability analysis to adaptive optimization algorithms. They use this analysis to hypothesize why adaptive optimizers converge to sharper minima compared to SGD, which has been associated with poorer generalization performance [1],[2]. The authors also introduce a new measure, which they term the Generalized Coherence Measure, where they also show a correlation between a lower generalized coherence measure and better test accuracy
The authors provide some experiments on vision tasks, where it is also observed that better test accuracy is associated with smaller maximum eigenvalue and lower sharpness.

**Strengths:**

The authors discuss an important topic on the dynamics of adaptive methods. Also, the paper is very easy to follow.

**Weaknesses:**

- Overall the paper is poorly written, and a section on related work is missing. Important related work such as [1] was not mentioned.

- I believe that Eq. (13), which is a key part of this paper, is simply wrong. One cannot decompose the eigenvalues of
$M = I - \eta P^{\star-1} H(\theta^{\star})$
as $1 - \eta \frac{\lambda_i}{p_i}$ because this requires $P^{\star-1} $ and $H (\theta^{\star})$ to be co-diagonalizable, which is generally not the case.

- The novelty of this paper is quite limited. Apart from introducing a new measure, which seems to correlate with the test accuracy, the authors mainly just confirm the connections that has already been observed previously (such as the hypothesized connection between lower sharpness and better generalization).

- Miscellaneous errors, such as: in line 83: Figure ?? not referenced correctly, in line 117: $\rho$ not defined, in Figure 2: only one minimum is shown...


[1] Cohen, Jeremy M., et al. "Adaptive gradient methods at the edge of stability." arXiv preprint arXiv:2207.14484 (2022).

**Questions:**

see weaknesses

---

### Official Review · Reviewer_wp95 · 2024-11-04

**Soundness:** 2
**Presentation:** 3
**Contribution:** 2
**Rating:** 5
**Confidence:** 3

**Summary:**

The paper provides a theoretical framework for studying the stability properties of adaptive optimization algorithms such as Adam and RMSProp. The key idea is to introduce a generalized coherence measure, quantifying the interaction between the preconditioner and the Hessian of the loss function. The analysis presents one justification for why adaptive may often converge to sharper minima, leading to worse generalization performance. The authors demonstrate how the proposed framework could be used to tune the hyperparameters for Adam. Empirically, the authors justify their framework on standard image classification tasks by showing the relationship between test accuracy, sharpness, and maximum eigenvalue of different optimizers.

**Strengths:**

- The paper is well-written, has clear motivation, and presents valuable contributions to the ICLR community. The background section is particularly thorough and accessible.
- The theoretical analysis appears correct and provides practical guidelines for tuning Adam's hyperparameters.

**Weaknesses:**

- While the linear stability analysis is valuable, the framework may not fully capture why adaptive optimization algorithms lead to larger generalization gaps. SGD and Adam exhibit different implicit biases and optimization trajectories, which could be the primary factors. What would happen if one replaces SGD on an Adam-trained network near convergence?
- Pan et al. [1] also show that Adam leads to sharp minima (they are more stable in sharp regions). Could the authors clarify how this analysis differs?
- Key assumptions are not empirically justified, particularly the assumption about preconditioner convergence to a constant at the training's end. It would be helpful to describe the limitations of the proposed framework.
- The empirical analysis feels limited, focusing mainly on image classification tasks. Validation on other domains (NLP, RL) would strengthen the claims. However, although this is one weakness of the paper, I did not put much weight on it. Several key details are missing to reproduce the experiments in the paper (e.g., how the authors chose the hyperparameters). I believe that properly describing these details is important for optimization (or analysis) works. I am willing to increase my score if these details are properly described.

[1] Zhou, Pan, et al. "Towards theoretically understanding why sgd generalizes better than adam in deep learning." Advances in Neural Information Processing Systems 33 (2020): 21285-21296.

**Questions:**

- How does the framework extend to other adaptive optimization methods beyond Adam and RMSProp?
- According to this analysis, why does Adam perform better on transformer-based architectures (e.g., language modeling)?
- (Minor) Figure number is missing in line 82.

---

### Official Review · Reviewer_an1a · 2024-11-10

**Soundness:** 3
**Presentation:** 3
**Contribution:** 2
**Rating:** 5
**Confidence:** 3

**Summary:**

This paper expands the scope of linear stability analysis to encompass adaptive optimization algorithms such as Adam and RMSProp. It establishes a theoretical framework that links the stability characteristics of these algorithms to the topography of the loss landscape. The paper introduces an innovative generalized coherence metric that assesses the interplay between the adaptive preconditioner and the Hessian matrix of the loss function. This metric yields both necessary and sufficient conditions for linear stability in the vicinity of stationary points. The study's results indicate that adaptive optimizers have the capacity to handle sharper minima but may suffer from inferior generalization when compared to conventional methods like SGD. The paper also provides actionable advice on hyperparameter tuning to address this potential drawback.

**Strengths:**

This paper presents an extensive theoretical framework that broadens the conventional stability analysis to include adaptive optimization algorithms, deepening our comprehension of their operational dynamics in the context of the loss surface's geometric properties.

It introduces a groundbreaking coherence measure that quantifies the relationship between the adaptive preconditioner and the Hessian matrix, shedding light on the prerequisites for linear stability and their impact on convergence patterns.

The study offers practical recommendations for hyperparameter tuning, designed to bolster the generalization capabilities of adaptive optimizers. This addresses the noted disparity in generalization performance when compared to stochastic gradient descent (SGD).

**Weaknesses:**

The contribution of this work appears limited, as it seems to build incrementally on existing stability analyses of SGD. The primary difference highlighted is that while the precondition matrix in SGD is an identity matrix, in the adaptive method, it is a diagonal matrix with positive values.

In the theoretical proof, the precondition matrix is assumed to be constant. However, in the experiments, when the model weights converge, the precondition matrix may change slowly rather than remaining stable. It is unclear whether these small changes would affect the validity of the proof provided in the paper.

Some details of the experiment are unclear, particularly concerning Figure 1 and Table 2. Were these results obtained by training from scratch? Were the experiments run multiple times to ensure consistency? In Figure 1, the results suggest that the method converges only during the initial phase of the training period, indicating that further tuning may be needed.

**Questions:**

Could the authors consider making their code publicly available to facilitate reproduction of the study?

---

### Meta-Review · Area_Chair_5C1J · 2024-12-20

**Metareview:**

This paper extends linear stability analysis to include adaptive optimization algorithms like Adam and RMSProp. It develops a theoretical model connecting the stability of these algorithms to the shape of the loss function's landscape. It also introduces a new index to measure how well the algorithm's adaptive preconditioner aligns with the curvature of the loss function. This index is used to determine stability around stationary points and offers insights into why adaptive methods may converge to sharper minima with poorer generalization. The theory is supported by experiments on benchmark datasets and architectures.

The reviewers appreciate that the presented theoretical framework can broaden conventional stability analysis to include adaptive optimization algorithms, deepening our comprehension of their operational dynamics in the context of the loss surface's geometric properties. They also appreciate the new coherence measure, which quantifies the relationship between the adaptive preconditioner and the Hessian matrix. They also find the paper clearly motivated and with a thorough and accessible background section.

Despite these strengths, the paper has issues that need to be addressed before it can be published. In particular, reviewer c15z believes the paper has significant technical flaws, with mathematical arguments that lack both rigor and clarity, and provides a detailed list of areas where these issues arise. Similarly, reviewer px69 believes that Eq. (13), which is a key part of this paper, is wrong. Reviewers an1a and px69 both ask for the author's clarification on the novelty of the contributions of the paper: they both believe the current contributions build incrementally on existing stability analyses of SGD. Reviewers an1a and wp95 also express concern about the assumption that the preconditioner matrix is constant in the theoretical proofs.

The authors decided not to respond to the feedback, which leaves the issues unresolved. Given the unresolved concerns, especially around errors and flaws in the theory part of the paper, the paper cannot be accepted in its current form. I encourage the authors to consider resubmitting their work after fixing the problems mentioned in this review cycle.

**Additional Comments On Reviewer Discussion:**

Reviewer c15z believes the paper has significant technical flaws, with mathematical arguments that lack both rigor and clarity, and provides a detailed list of areas where these issues arise. Similarly, reviewer px69 believes that Eq. (13), which is a key part of this paper, is wrong. Reviewers an1a and px69 both ask for the author's clarification on the novelty of the contributions of the paper: they both believe the current contributions build incrementally on existing stability analyses of SGD. Reviewers an1a and wp95 also express concern about the assumption that the preconditioner matrix is constant in the theoretical proofs.

---

### Decision · Program_Chairs · 2025-01-22

Reject